# Antihypertensive, cardio- and neuro-protective effects of *Tenebrio molitor* (Coleoptera: Tenebrionidae) defatted larvae in spontaneously hypertensive rats

Federica Pessina[1] ʘ, Maria Frosini[2] ʘ *, Paola Marcolongo[1], Fabio Fusi[3], Simona Saponara[2], Alessandra Gamberucci[1], Massimo Valoti[2], Daniela Giustarini[3], Paolo Fiorenzani[4], Beatrice Gorelli[2], Valeria Francardi[5], Maurizio Botta[3,6], Elena Dreassi[3]

1 Dipartimento di Medicina Molecolare e dello Sviluppo, Università degli Studi di Siena, Siena, Italy, 2 Dipartimento di Scienze della Vita, Università degli Studi di Siena, Siena, Italy, 3 Dipartimento di Biotecnologie, Chimica e Farmacia, Università degli Studi di Siena, Siena, Italy, 4 Dipartimento di Scienze Mediche, Chirurgiche e Neuroscienze, Università degli Studi di Siena, Siena, Italy, 5 Consiglio per la Ricerca in Agricoltura e l'Analisi dell'Economia Agraria, Centro di ricerca Difesa e Certificazione (CREA-DC), Impruneta (Firenze), Italy, 6 Lead Discovery Siena Srl, Castelnuovo Berardenga, Siena, Italy

ʘ These authors contributed equally to this work.
* maria.frosini@unisi.it

**Data Availability Statement:** All relevant data are within the manuscript and its Supporting Information files.

## Abstract

In pre-hypertension, moderate control of blood pressure (BP) can be obtained by a nutritional approach. The effects of a diet enriched with defatted larvae of the mealworm *Tenebrio molitor* (Coleoptera: Tenebrionidae) (TM) endowed with ACE inhibitory activity was studied in both spontaneously hypertensive rats (SHR) and in the age-matched normotensive Wistar Kyoto strain. These were fed for 4 weeks with standard laboratory rodent chow supplemented with or without TM or captopril. In SHR, the TM diet caused a significant reduction in BP, heart rate and coronary perfusion pressure, as well as an increase in red blood cell glutathione/glutathione disulphide ratio. Rat brain slices of SHR were more resistant to oxidative stress and contained lower levels of inflammatory cytokines, while vascular and liver enzyme-activities were not affected. These results suggest that TM can be considered a new functional food that can lower BP *in vivo* and thus control cardiovascular-associated risk factors such as hypertension.

## Introduction

Hypertension is the main risk factor for diseases such as coronary artery disease, stroke, heart failure, atrial fibrillation and dementia. The incidence of cardiovascular diseases and related deaths is expected to rise worldwide. The Seventh Joint National Committee (JNC7) report recently proposed the concept of prehypertension (systolic blood pressure 120–139 mmHg and/or diastolic blood pressure 80–89 mmHg), which can progress to hypertension, impairing cognitive function, increasing left ventricular mass and causing end-stage renal disease and

**Funding:** This research was carried out under the project "INSECT CARD" funded by the Tuscan Regional Administration's Call for Research Projects in the Nutraceutical Industry. Executive Decrees No. 4741 of 21 October 2014 and No. 5418 of 17 November 2014, PI Valeria Francardi. Lead Discovery Siena Srl provided support in the form of research materials. The funders did not play any role in the study design, data collection and analysis, decision to publish, or preparation of the manuscript.

**Competing interests:** The authors have read the journal's policies and have the following competing interests to declare: Lead Discovery Siena Srl provided support in the form of research materials. MB is the CEO of Lead Discovery Siena Srl, but did not receive salary from Lead Discovery Siena Srl. There are no patents, products in development or marketed products associated with this research to declare. This does not alter our adherence to PLOS ONE policies on sharing data and materials.

arteriosclerosis [1]. Instead of drug therapy, dietary approaches to stop hypertension (DASH) and the Mediterranean diet associated with appropriate lifestyle could be ideal measures to implement in cases of prehypertension [2], particularly in patients with concomitant cardiovascular risk factors. Since diet plays a crucial role in maintaining healthy blood pressure, health-conscious consumers seek functional foods to reduce cardiovascular risk factors and improve vascular health.

Edible insects were recently proposed as novel functional food by virtue of their antitumor, antiobesity, antimicrobial, antioxidant and antihypertensive properties [3]. For example, the protein fraction from gastrointestinal [4] or alcalase enzyme hydrolysis [5] of larvae of *Tenebrio molitor* (Coleoptera Tenebrionidae) (TM) shows ACE inhibitory activity. Considering the ease of breeding TM larvae, they may be used as a functional ingredient in fortified food and/or as a dietary supplement to lower blood pressure for cardiovascular prevention in pre-hypertensive patients. The aim of the present study was to investigate the effects of the protein obtained from the larval stage of TM, *in vivo* and *ex-vivo*, in spontaneously hypertensive rats (SHRs) and in age-matched rats of the normotensive Wistar Kyoto strain (WKY). The ACE inhibitor captopril was used in parallel as a positive control. The diet supplemented with TM was found to limit blood pressure increase and to protect the brain against oxidative stress, thus indicating TM as a possible functional food.

## Materials and methods

### T. molitor larva meal

Larvae of TM were obtained by mass rearing at CREA-DC (Italy), as previously described [4]. Specimens were kept in plastic boxes in a climatic room (24˚C ±2 and 70%±5 relative humidity) in the dark. The breeding diet was composed of yeast (5%), wheat flour (47.5%) and oat flour (47.5%) plus carrot pieces as source of water. The larvae were not starved before being killed. They were dried at 50˚C under vacuum and ground using a food processor before fat extraction with ethanol (99.5%) [6]. The solvent was then removed under vacuum at 25˚C for 24 h and the larva meal used for feed preparation. The nutritional value and other details of defatted TM are reported in **S1 File**. Mucedola Srl (Italy) prepared the standard diet (4RF25, for composition see www.mucedola.it) supplemented with larva protein or with captopril (see **S1 Table**).

### Animals and ethical statement

The procedures used complied with European legislation on the use and care of laboratory animals (EU Directive 2010/63) and National Institutes of Health guidelines, and were approved by the Italian Ministry for Health (185/2015PR).

Nine-week old male spontaneously hypertensive rats (SHRs, n = 24) and age-matched WKY rats as controls (n = 18) were purchased from Charles River (Italy, www.criver.com). Two to four rats per cage were housed in a room maintained at constant temperature (23–24˚C) and humidity (50–60%), on a 12 h light/dark cycle, with free access to food and water.

### Experimental design

At ten weeks of age, when hypertension develops [7], hypertensive and WKY rats were randomly divided into three groups, and fed different diets for four weeks as indicated in **Fig 1**. At the end of this period, micturition frequency was assessed: all rats were then anaesthetized (for details, see **S2 File**), blood samples were collected and heart, aorta, liver, kidney and brain were quickly explanted.

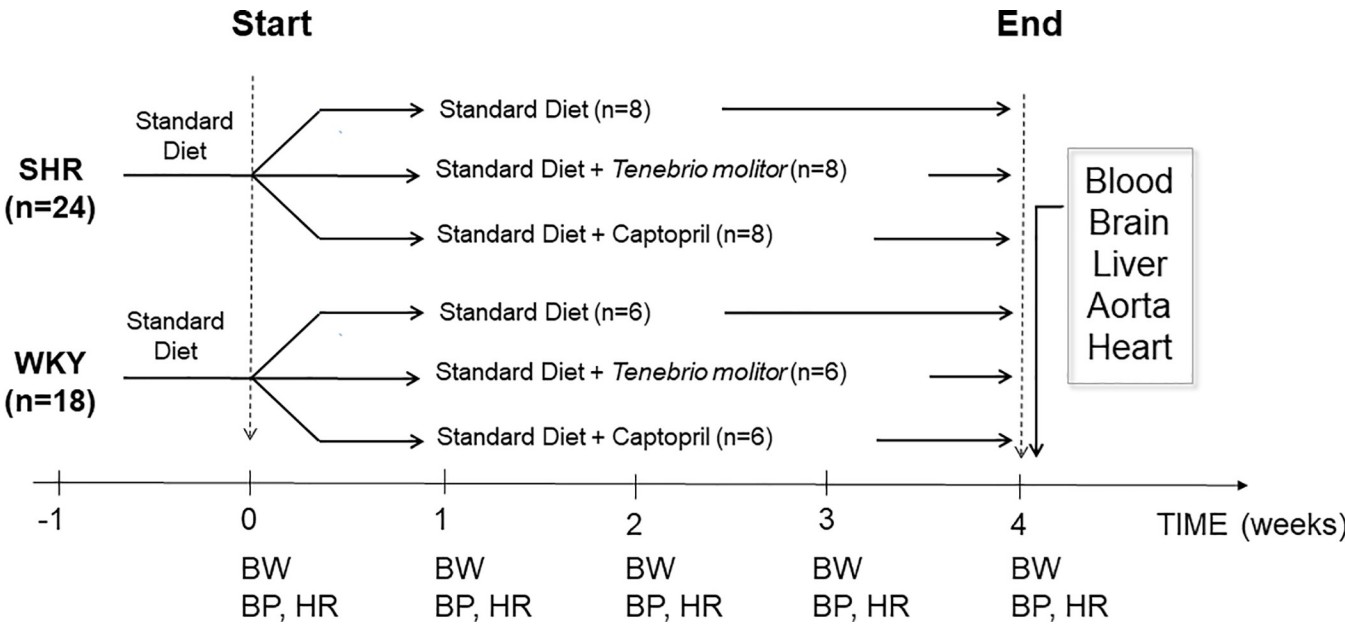

**Fig 1. Overview of experimental design.** After a one-week housing period, both Wistar Kyoto (WKY) and spontaneously hypertensive rats (SHRs) were randomly assigned to standard diet (SD), SD supplemented with *Tenebrio molitor* or captopril. Both strains were fed for 4 weeks, when body weight (BW), blood pressure (BP) and heart rate (HR) were measured on a weekly basis. After four weeks, the rats were sacrificed and their blood, heart, aorta, liver and brain harvested.

### Doses

Since the diets were supplemented with 4.5% larva protein or 100 mg/kg captopril, and on the basis of rat body weight (see **S2 Table**) and on average daily food intake (18.7 g, see **S3 Table**), a daily dose of ~2.9 g/kg of larva protein and ~8 mg/kg body weight captopril was estimated. Captopril reduces blood pressure [8, 9] and inhibits plasma ACE activity [10] in SHRs. It is used to treat mild to moderate hypertension in humans [11].

### Blood pressure and heart rate measurement

Systolic blood pressure was measured weekly by the non-invasive "tail-cuff" method [12]. After warming the body of the rat to 37°C for 5 min, the average of three consecutive blood pressure readings, taken between 9am and 12am, was considered. In parallel, heart rate, i.e. heart beats per minute, was also recorded.

### Plasma and brain ACE and inhibitory activity

ACE was detected in plasma and brain homogenates by measuring the amount of hyppuric acid cleaved by ACE from the substrate N-hyppuryl-L-histidyl-L-leucine. ACE inhibitory activity was assessed according to a previous protocol [13], based on quantification of the hyppuric acid released in samples by addition of a known amount of ACE. Finally, hyppuric acid was determined by RP-HPLC according to Wang et al. [13]. Details are reported in **S3 File**.

### RBCs and plasma thiols

Blood (~0.5 ml) was collected in tubes containing 50 µl N-ethylmaleimide solution, 310 mM. After 1 min, blood was centrifuged at 10,000xg for 20s and plasma was collected. The pellet was washed twice with saline and the purified RBCs were deproteinized by addition of 1

volume 15% (w/v) trichloroacetic acid. Glutathione (GSH) and glutathione disulphide (GSSG) were measured in the supernatant by UV-Vis HPLC and the GSH recycling method, respectively, while hemoglobin concentration was measured spectrophotometrically (Jasco V-530 instrument, Jasco Europe Srl, Italy) [14].

Total thiols were measured in plasma samples by HPLC (Agilent series 1100 instrument, Agilent Technologies, Italy) by labelling SH groups with the fluorescent probe monobromobimane (Calbiochem) after reduction of disulphide bridges with dithiothreitol [14].

## Liver cytochrome P450 and $b_5$ content, NADPH-cytochrome P450 reductase activity

Liver microsomes were prepared according to [15] (details in **S4 File**). Cytochrome P450 (CYP) and cytochrome $b_5$ contents were measured from the CO-difference spectra of the microsomes at ΔA 450–490 nm (ε: 91 mM) and ΔA 424–490 nm (ε: 112 mM), respectively [15]. NADPH-cytochrome P450 reductase activity was measured by following cytochrome c reduction at 550 nm [15].

## Brain IL-6, IL-1β and TNF-α

IL-6 (Millipore®), IL-1β (Thermo Fisher Scientific) and TNF-α (Boster PicoKine[TM] ELISA) were measured according to the manufacturer's instructions in brain homogenates prepared as detailed in **S5 File**. Values were normalized to the protein content of the sample.

## Ex-vivo functional studies

**Aorta ring mechanical activity.** Rings were prepared as previously described [16]. Relaxation of phenylephrine-contracted rings to either acetylcholine or sodium nitroprusside was taken as an index of endothelial function and muscle sensitivity to NO, respectively (more details in **S6 File**).

**Isolated heart preparation and perfusion.** Spontaneously beating hearts were rapidly explanted and mounted on a Langendorff apparatus for retrograde perfusion with physiological salt solution *via* the aorta at a constant flow rate of 10 ml/min, as described elsewhere [17] (**S7 File**).

Heart contractility was measured as left ventricle developed pressure (LVDP = left ventricular systolic pressure minus left ventricular diastolic pressure) by means of a latex balloon, inserted into the left ventricle *via* the mitral valve and connected to a pressure transducer (BLPR, WPI, Berlin, Germany). Coronary perfusion pressure was recorded by a pressure transducer in the inflow line. Heart rate was calculated from LVDP. LVDP and coronary perfusion pressure were recorded with a digital PowerLab data acquisition system (PowerLab 8/30; ADInstruments, Australia) and analysed with Chart Pro for Windows software (PowerLab; ADInstruments, Australia).

**Brain susceptibility to oxidative stress-mediated injury.** Brain cortical slices were prepared according to [18] (see also **S8 File**). Briefly, the two brain hemispheres were divided: one was homogenised and used for cytokine assessment (**S5 File**); the other was cut into 400-μm slices. These were exposed (OS) or not exposed (basal) to 10 mM $H_2O_2$ for 1 h. Then the colorimetric MTT assay was performed to assess tissue viability [19]. Slice tissue edema, an additional index of tissue viability [20], was calculated using the formula [18]:

$$\text{tissue edema} = (\text{wet weight} - \text{dry weight})(\text{dry weight})^{-1}.$$

## Statistical analysis

Results are reported as mean ± SEM. Statistical significance was assessed by ANOVA and Bonferroni post-test, Student t test or one-sample t test, as appropriate (GraphPad Prism version 5.04, GraphPad Software Inc., San Diego, CA). In all comparisons, statistical significance (*P*) was set at 0.05.

## Results

### Effect of TM on body weight and food intake

No changes in skin, fur, eyes, mucous membranes or salivation were observed in any of the treated groups, nor were there tremors or death. The TM and captopril diets did not affect growth rate, which was comparable to that of rats fed the standard diet (final weight gain about 43 g, see **S2 Table**). Similar results were obtained for food intake, which ranged from 17.7±0.1 to 19.6±0.6 g/rat/day (see **S3 Table**).

### Effects of TM on cardiovascular mechanics

Basal blood pressure was much higher in hypertensive than in WKY rats (Fig 2, panel a). After four weeks, only the former showed a significant increase, which was reduced by TM and captopril supplementation. No effect was observed in WKY rats. Noticeably, in SHRs, captopril maintained BP around basal values (150.0±2.8 mmHg) already from the first week of treatment, while TM took 4 weeks to regain the basal values of 146.2±6.5 mmHg. Similar results were observed for heart rate (Fig 2, panel b).

Vascular endothelial function and smooth muscle sensitivity to $\alpha_1$ adrenergic agonists as well as to NO were also assessed. Aorta rings from hypertensive and WKY rats showed similar responses to phenylephrine (**S1 Fig**). Captopril did not modify this response. On the contrary, TM supplementation generally increased the sensitivity of the preparations to phenylephrine, reaching significance in SHR rats. Rings from untreated SHRs stimulated by phenylephrine showed lower endothelium-dependent vasodilator responses to ACh than matched WKY rats ($E_{max}$ = 61.2±4.0%, vs. 86.1±2%, respectively; P<0.05) (Fig 2, panel c). This difference disappeared when rings were pre-incubated with 5 μM indomethacin. Captopril ($E_{ma}$x 74.7±5.0%, P>0.05 vs. WKY), but not TM ($E_{max}$ 53.2±5.5%, P<0.05 vs. WKY), produced a significant increase in ACh-induced relaxation of SHR rings. The response to SNP was similar in WKY rats and untreated SHRs, as well as in SHRs treated with captopril or TM (**S1 Fig**) Finally, neither captopril nor TM modified the response of WKY rings to ACh or SNP (**S1 Fig**).

Regarding heart function, no type of diet fed to WKY or SHRs affected LVDP or heart rate as evaluated by the Langendorff perfused heart model (**S2 Fig**). SHRs fed the standard diet showed higher coronary perfusion pressure than did the matched normotensive strain, with both captopril- and TM -supplemented diets bringing this parameter back to control values (Fig 2, panel d).

### Effects of TM on red blood cells and on plasma thiol and disulfide levels

Red blood cell glutathione (GSH) levels were similar in normotensive and hypertensive rats and were not affected by TM or captopril diets (Fig 3, panel a). TM supplementation significantly increased the GSH/GSSG ratio in SHRs (Fig 3, panel b), suggesting that in this group the diet may have been responsible for a decrease in oxidative stress. In standard diet-fed rats, plasma total homocysteine (tHcys) was significantly lower in hypertensive than in matched normotensive rats (Fig 3, panel c). This difference was partially reduced by TM and completely reversed by the captopril diet. Finally, in SHRs, TM caused a significant increase in the total

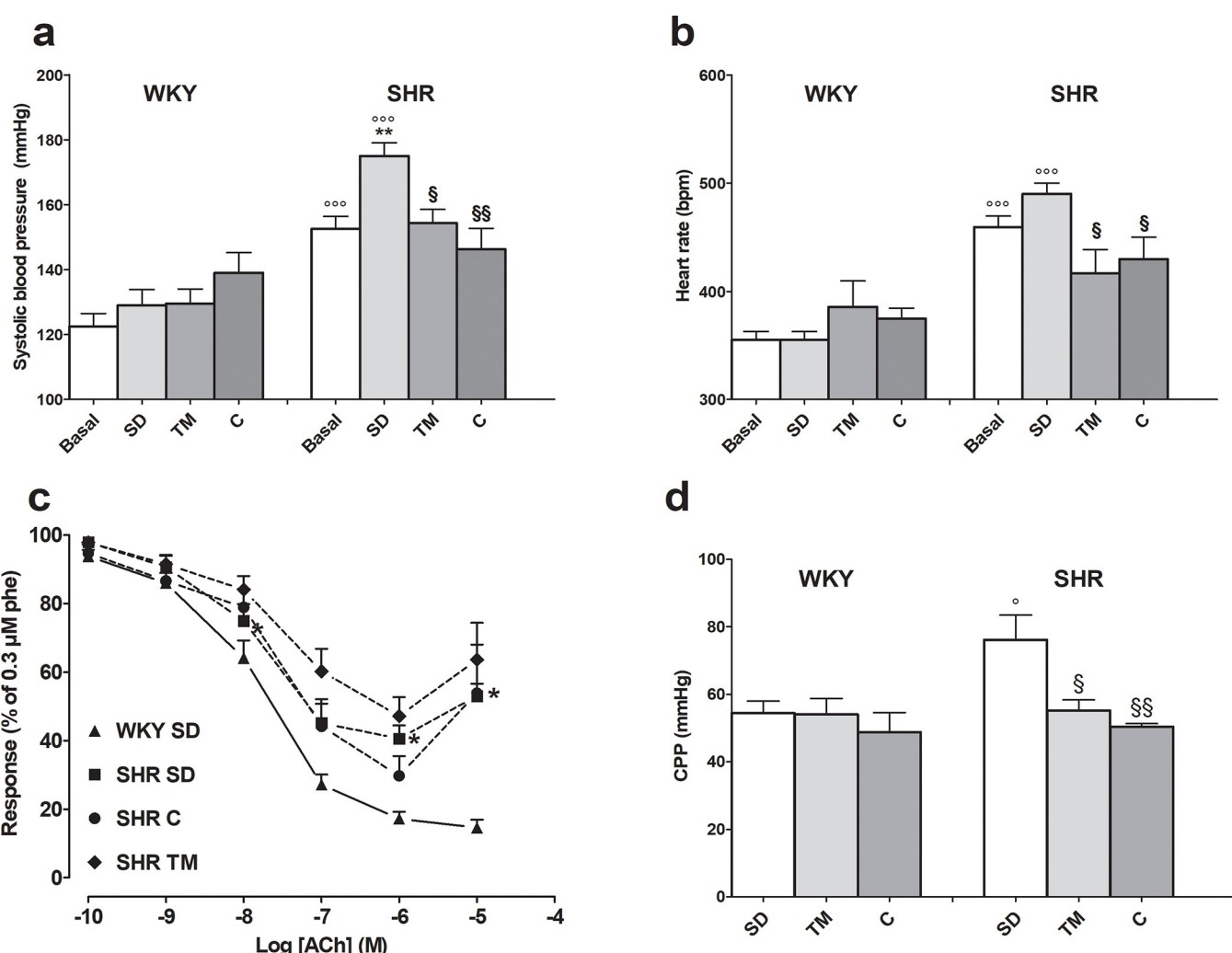

**Fig 2. Effects of feeding hypertensive and WKY rats for 4 weeks with standard laboratory rodent chow (SD) or SD supplemented with *Tenebrio molitor* (TM) or captopril (C) on cardiovascular systems.** Panel a and b: systolic blood pressure and heart rate in hypertensive and WKY rats. Basal: values recorded at the beginning of the 4 weeks of treatment. Data is reported as mean±SEM. **P<0.01 vs SHR basal; §P<0.05, §§P<0.01 vs SHR SD (ANOVA followed by Bonferroni post test); °°°P<0.01 vs WKY, same diet (Student t test, unpaired samples). Panel c: vascular endothelial function, concentration-response curve to acetylcholine (ACh) of endothelium-intact rat aorta rings pre-contracted with 0.3 μM phenylephrine. On the ordinate scale, response is reported as percentage of contraction with respect to that induced by phenylephrine (phe, 100%). Values are reported as mean±SEM. *P<0.05 SHR SD vs WKY SD (Student's t test, unpaired samples). Panel d: coronary perfusion pressure (CPP) on isolated heart. Data reported as mean±SEM. §P<0.05, §§P<0.01 vs SHR SD (ANOVA followed by Bonferroni post test); °P<0.05 vs WKY same diet (Student t test, unpaired samples).

level of other physiological plasma thiols (i.e. cysteine, cysteinylglycine, γ-glutamylcysteine and GSH) compared to matched normotensive rats (see **S4 Table**).

## Effect of TM on liver CYP450, cytochrome b$_5$ levels and NADPH-cytochrome P450 reductase activity

Liver CYP450 and cytochrome b$_5$ content, as well as NADPH-cytochrome P450 reductase activity were unaffected by TM and captopril treatment (see **S5 Table**). Likewise, liver weight gain did not change significantly after 4 weeks on standard diet or diets supplemented with TM or captopril, averaging the value of 10.4 g (lower 95% conf. limit 9.7 g; upper 95% conf. limit 11.2 g).

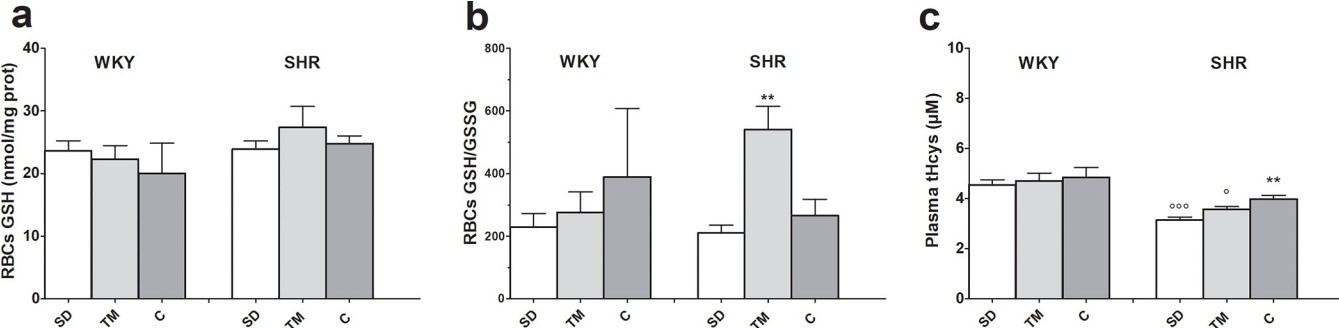

**Fig 3.** Effects of feeding hypertensive and WKY rats for 4 weeks with standard laboratory rodent chow (SD) or SD supplemented with *Tenebrio molitor* (TM) or captopril (C) on red blood cell GSH levels (panel a), GSH/GSSG ratio (panel b) and plasma total homocysteine (tHcys, panel c). Data reported as mean ±SEM. **$P<0.01$ vs SHR SD; °$P<0.05$, °°°$P<0.001$ vs WKY, same diet (ANOVA followed by Bonferroni post test).

### Effect of TM on plasma and brain ACE content and inhibitory activity

As shown in Fig 4, panel a, plasma ACE levels were comparable in all experimental groups, while in brain its content was about 30 fold higher than in plasma (Fig 4 panel c). Furthermore, both TM—and captopril-fed SHRs showed higher brain tissue ACE levels than rats fed the standard diet, though this difference did not reach statistical significance. Brain enzyme inhibitory activity was not affected by the different diets (Fig 4, panel d); on the contrary, in plasma it was significantly higher in SHRs fed with TM and captopril than in rats fed the standard diet (Fig 4, panel b).

### Brain IL-1, IL-6 and TNF-α content

Diet supplementation had no effect on brain cytokine levels in WKY rats, whereas SHRs fed the standard diet showed significantly higher brain IL-1ß and TNF-α, but not IL-6 content, than the matched normotensive strain (Fig 5 panel a and b and **S3 Fig**). This effect was reversed by TM supplementation, while captopril only reduced IL-1ß. Finally, IL-6 was not affected by diet in SHRs (**S3 Fig**).

### Effects of TM on brain cortical slice susceptibility to oxidative stress-mediated injury

Hydrogen peroxide challenge caused a significant loss in brain tissue viability in both groups of rats fed the standard diet (Fig 5, panel c). Interestingly, TM treatment protected the viability of slices from this injury in both rat strains. Captopril was effective in SHRs, but not in WKY rats. Tissue edema, taken as an additional index of viability, mostly mirrored the results of the MTT assay (Fig 5, panel d). Indeed, oxidative stress induced a significant increase in tissue water gain in both groups of rats. Brain cortical slices of both groups of rats fed with TM had a significantly lower water content than the same strain fed the standard diet. In addition, captopril reduced tissue edema, although it was only effective in WKY rats.

### Discussion

The present results highlight interesting *in vivo* antihypertensive, cardio- and neuro-protective effects of protein hydrolysates derived from the larval stage of TM that could be useful for treating borderline blood pressure values in humans. Dietary supplementation with TM appeared to be safe, as no significant effects on food intake, weight gain or physical attributes of the animals were recorded, as already reported by [21]. Similar effects were obtained with

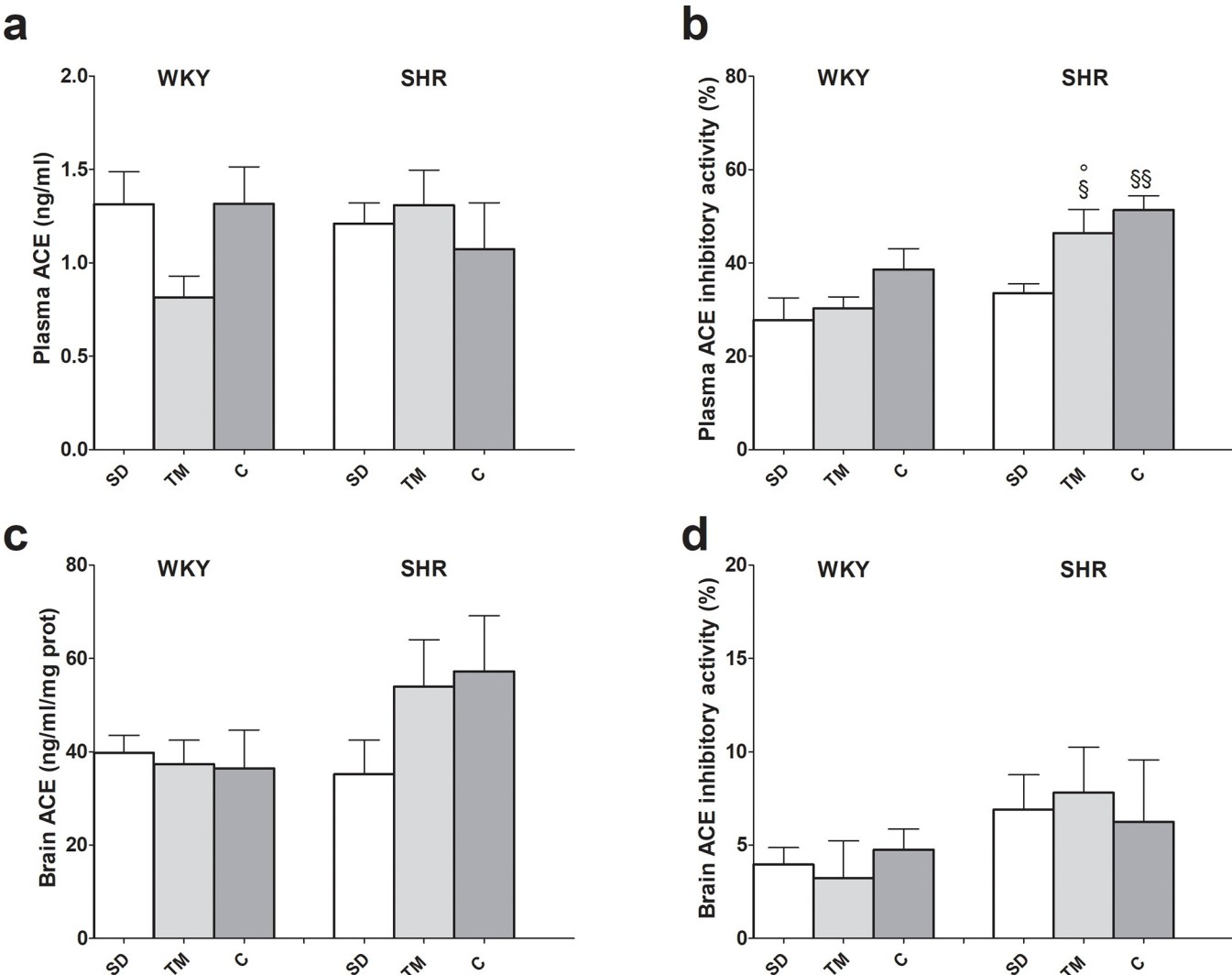

**Fig 4. Effects of feeding hypertensive and WKY rats for 4 weeks with standard laboratory rodent chow (SD) or SD supplemented with *Tenebrio molitor* (TM) or captopril (C) on plasma and brain ACE content and inhibitory activity.** Values are reported as mean ± SEM. §P<0.05, §§ P<0.01 vs SHR SD (ANOVA and Bonferroni post test); ° P<0.05 vs WKY same diet (Student t test, unpaired samples).

captopril, although this drug did not counteract oxidative stress in brain slices. More importantly, TM did not affect liver microsomal CYP content or the activity of the enzymes responsible for their kinetics, suggesting that TM intake is without drug-interaction effects. Modulation of CYP-dependent enzyme activities by nutraceuticals could in fact alter the pharmacokinetic properties and/or toxicity of a concomitantly administered drug [22]. The fact that TM did not inhibit and/or induce CYPs linked to drug metabolism sustains the possibility of co-therapies without risk of an increase in drug-related adverse reactions.

A feature of particular relevance shown by TM was its ability to prevent the marked decrease in blood pressure and heart rate observed in SHRs, known to be associated with vascular and heart protection [23]. The negative chronotropic effect was not, however, observed in *ex vivo* Langendorff isolated hearts. Absence of the physiological humoral background could explain the discrepancy existing between *in vivo* and *ex vivo* chronotropic effects of the TM-enriched diet. Hypertensive rats fed the standard diet showed higher coronary perfusion

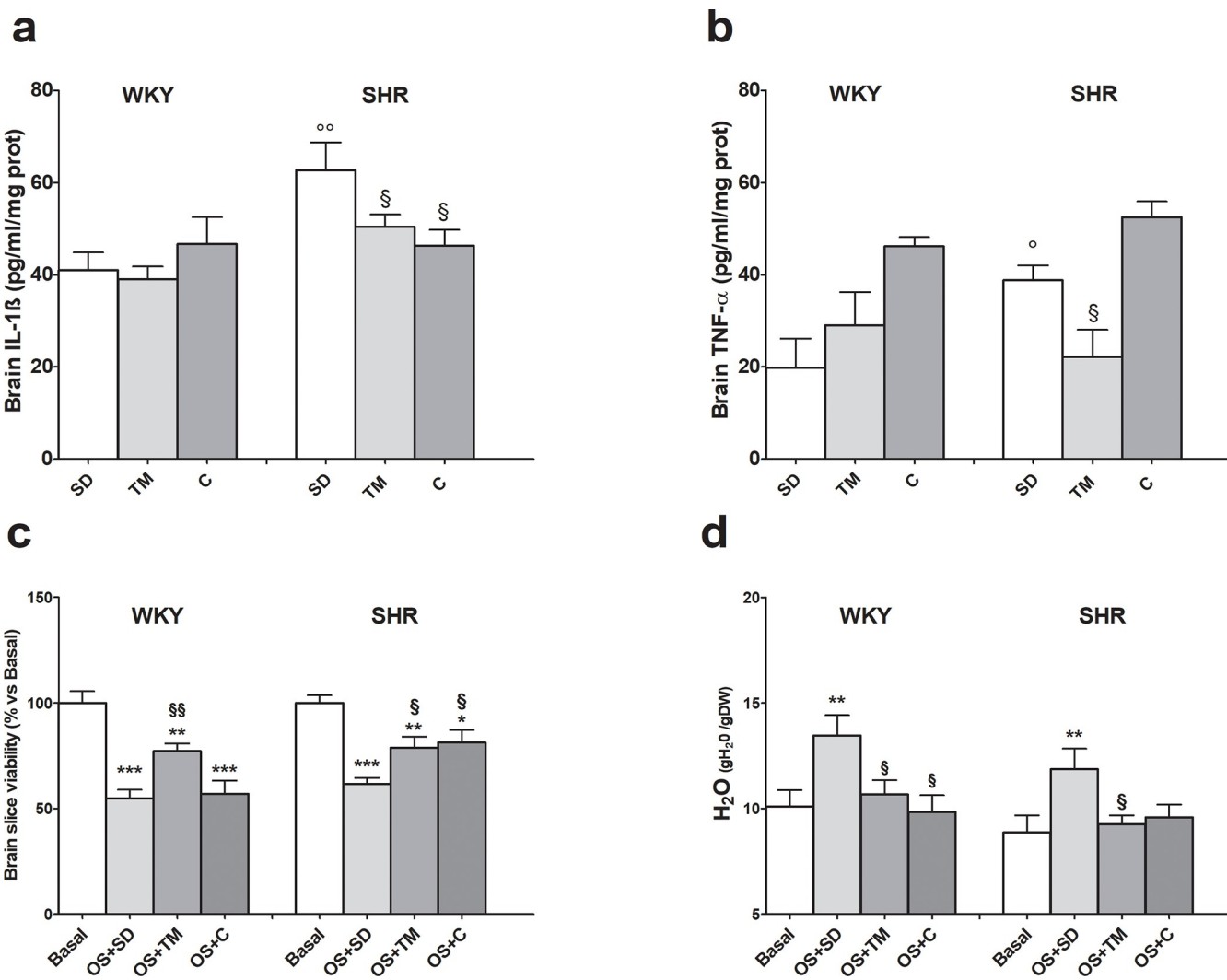

**Fig 5.** Effects of feeding hypertensive and WKY rats for 4 weeks with standard laboratory rodent chow (SD) or SD supplemented with *Tenebrio molitor* (TM) or captopril (C) on IL-1 ß (panel a) and TNF-α (panel b) brain content and on susceptibility of brain cortical slices to hydrogen peroxide-induced injury (10 mM for 1 h, oxidative stress OS) assessed as tissue viability (panel c) and edema (panel d). Values are reported as mean ± SEM. Brain cytokines: §$P < 0.05$ vs SHR SD (ANOVA and Bonferroni post test); ˚$P < 0.05$, ˚˚$P < 0.01$ vs WKY same diet (Student t test, unpaired samples). Brain slices: *$P < 0.05$, **$P < 0.01$, ***$P < 0.001$ vs basal; §$P < 0.05$, §§$P < 0.01$, vs OS+SD (ANOVA and Bonferroni post test).

pressure than the normotensive strain, as previously described [24]. Interestingly, the TM-enriched diet reduced coronary perfusion pressure to the same extent as captopril in WKY rats, bringing its value back to control level without changing heart function. Finally, the ACE inhibitory activity recorded in plasma of SHRs (standard diet, TM and captopril) was significantly correlated with the decrease in blood pressure recorded at the end of the 4 weeks ($r^2 = 0.9975$). This suggests that the mechanism of action underlying the antihypertensive effect of TM in lowering blood pressure can be ascribed to TM-induced inhibition of ACE.

As weight gain in the TM diet group of SHRs seemed less than that of the SHR control group, it can be postulated that a slight reduction in fat mass could contribute to the hypotensive effect exerted by TM. Indeed, Seo and colleagues reported an antiobesity effect of TM supplementation in mice fed a high fat diet [25]. Although the increase in blood pressure in the SHR group is mostly genetically driven, we cannot exclude that a slight reduction in body fat

might be involved. If so, TM supplements for treating pre-hypertension are even more interesting and warrant further investigation.

In hypertension, oxidative stress plays a critical role in the molecular mechanisms associated with cardiovascular and renal injury. Hypertension itself can contribute to oxidative stress [26]. The GSH/GSSG ratio in red blood cells is a biomarker of oxidative stress. Neither GSH nor GSH/GSSG values changed in SHRs, in line with a previous report [27], indicating a good blood antioxidant barrier against the burst of radicals associated with hypertension. Although the captopril diet lowered blood pressure, it did not modify GSH-GSSG balance, as already reported in humans [28]. In contrast, the TM diet increased the GSH/GSSG ratio, suggesting that its antihypertensive effect could be linked to the reduction in GSSG in red blood cells or *vice versa*.

A close relation exists between intracellular and extracellular thiol pools [29]. The low molecular mass Hcys in plasma is also considered an independent risk factor in cardiovascular disease as it causes endothelial dysfunction [30]. However according to other reports, SHRs have lower Hcys levels than WKY rats [31], suggesting that Hcys *per se* is not linked to the development of hypertension in this strain. Paradoxically, the TM diet partially, and the captopril diet completely restored Hcys levels to control values, once more demonstrating their potential to restore physiological balance in hypertensive animals.

Regression of structural and/or functional alterations of the vascular endothelium occurring in hypertension is an important target of antihypertensive therapy. In this study, impaired relaxation of aorta rings of SHRs was observed in response to ACh, in line with reports in the literature [32]. Among other things, this may result from impaired NO synthesis [33] and/or an increase in superoxide anions, as well as various contracting factors released by a dysfunctional endothelium. Partial recovery of response to ACh was recorded in SHRs treated with captopril, but not in those treated with TM. This can be ascribed to a protective effect of the drug towards endothelial function, rather than to improved sensitivity of vascular smooth muscle to NO. Indeed, no differences in the vasodilator capacity of the NO donor SNP were detected in endothelium-denuded rings of the different groups, indicating that TM did not improve vascular function, at least in *ex vitro* assays. Furthermore, while captopril did not affect the sensitivity of SHRs vascular smooth muscle to phenylephrine-induced stimulation, TM increased it, thus ruling out a possible involvement of the sympathetic system in its systemic, beneficial activity. In fact, TM significantly lowered blood pressure after 3–4 weeks of treatment, as did captopril after only one week: therefore, it would be worth investigating whether vascular function might also be improved by extending TM treatment.

A hyperactive brain renin-angiotensin system, oxidative stress and neuroinflammation in brainstem cardiovascular centres and other brain regions increase sympathetic activity in hypertension [34]. The pro-inflammatory cytokines TNF-α, IL-1β and IL-6 act as neuromodulators in the paraventricular nucleus (PVN) of the hypothalamus by stimulating production of cytotoxic reactive oxygen species that further propagate the sympatho-excitatory effect [35]. Standard-diet-fed SHRs showed significantly higher IL-1ß and TNF-α brain content than the matched normotensive group, as previously described [36]. IL-6, on the contrary, was similar in hypertensive and WKY rats, in contrast with what was previously observed at PVN level [37]. When considering the present results, however, it cannot be ruled out that IL-6 is indeed increased in discrete areas such as the PNV, but this could not be detected measuring whole-brain cytokine content. Interestingly, the TM-enriched diet restored cytokine content to basal values, unlike captopril which was effective only towards IL-1ß. This observation indicates that the antihypertensive activity of TM can at least partly be ascribed to its anti-inflammatory effect.

Both peripheral and brain renin-angiotensin systems play a fundamental role in hypertension, hence ACE inhibitors are currently used to treat it. Brain ACE content was unchanged in

WKY rats, while it seemed elevated in SHRs on the TM and captopril diets, although the increase did not reach statistical significance. However, chronic administration of captopril at high doses increases ACE in some brain area of SHRs, owing to suppression of inhibitory feedback of angiotensin II on ACE synthesis [38]. The apparent lack of activity of TM and captopril on brain ACE content may be explained by the low dose of captopril, which may only partly affect ACE levels. Moreover, since the enzyme was measured in whole brain, the possibility that ACE might reach higher values in specific areas cannot be ruled out.

Chronic hypertension is accompanied by brain damage caused by a hypoxic/ischemic mechanisms, in which oxidative stress plays a pivotal role [39]. High blood pressure is also related to milder, subtle and chronic forms of brain damage, especially those concerning cognitive function. This is why we assessed the susceptibility of the brains of rats fed with TM or captopril to oxidative-stress-mediated injury. Tissue viability was sharply improved in SHRs fed with TM and captopril diets. Among possible mechanisms of neuroprotection, ACE inhibition leading to an increase in brain bradykinin, which in turn reduces formation of reactive oxygen species [40], can be hypothesized. Neuroprotection against oxidative stress can also be ascribed to an increased GSH/GSSG ratio and a decrease in pro-inflammatory cytokines, which were more pronounced in TM- than in captopril-treated rats, the former being more resistant than the latter, even in the WKY strain. The observation that acute injection of captopril up to 50 mg/kg does not reduce brain infarct area in rats subjected to middle cerebral artery occlusion [41] supports this hypothesis.

In conclusion, a diet enriched with protein derived from the larval stage of the mealworm *Tenebrio molitor* exerts antihypertensive, cardio- and neuro-protective effects. Since TM inhibits intrinsic blood coagulation pathways [42] and has antiobesity properties [25], it can be considered a multi-target functional food. TM larvae are already produced on a commercial scale, are inexpensive, nutritious [43] and devoid of toxicity [21], as well as being easy to raise and having less environmental impact than other solutions [44]. Taken together, all these characteristics make TM an optimal candidate for the development of a multi-target functional food for non-pharmacological treatment of prehypertension or mild hypertension in the framework of life-style advice. Since some of its effects are not shared by captopril, TM may also have added value in preventing hypertension and mitigating other severe cardiovascular risk factors. These are good prerequisites for a pharmacological characterization of the bioactive peptide components of TM with a view to treating pre-hypertension, a disease with high social impact.

## Supporting information

**S1 File. Defatting of T. molitor larvae.** Defatted larvae used in the paper were obtained extracting twice with ethanol (99.5%) for 1 h as reported by Zhao et al. [45]. The extraction was accomplished using a solvent to material ratio of 5 mL/g, at 40°C for 1 h. For the dry matter determination 3 g exactly weighed ground larvae were dried overnight at 104°C [46]. Fat content was determined by Soxhlet extraction method using petroleum spirit for six hours starting from 5 g of sample. The solvent was removed at 40° under rotary evaporator and extracted fat weighted after the flask reaching constant weigh after heating at 104°C. Crude protein were determined with the Kjeldahl method using a protein-to-nitrogen conversion factor of 6.25 [2]. Ash were determined by weighting after incineration of 2 g of sample at 550°C for 3 h and cooling in a desiccator [2]. Each assay was run at least in triplicate. The analysis gave in the starting material dry matter (95.8±0.08%), fat content (32.7±0.84%), crude proteins (50.8±0.72%) and ash (4.8±0.09%), extracted fat (33.8±0.92%) and crude protein content in the final product (77.5±0.59%). These values were superimposable to those of Zhao et al.

[45]. Nutritional value of TM protein as well as the *in vitro* digestibility was already accomplished [47]. Proteins extracted as water-soluble fraction (supernatant) or as water-insoluble fraction (pellet) were characterized by SD-PAGE and LC-MS/MS [48]. The most abundant in the supernatant were hemolymph protein (~12 kDa), alpha-amylase (~50 kDa, a putative allergen), and muscle proteins (*e.g.* actin 30–50 kDa) in the pellet fraction. Also the amino acid pattern and the essential amino acid index were already described [49] and [doi 10.3844/ajabssp.2009.319.331], showing that TM contain all the essential amino acid in the amount necessary to human [50]. The sum of total amount of amino acids (TAA) per g crude protein of TM was 910 mg/g, while the calculated essential amino acid index (EAAI) was slightly higher than that of soybean, but lower than that of casein. Moreover, the protein digestibility-corrected amino acid score (PDCAAS) of defatted TM was 0.80, further confirming that it has the ability to meet human dietary requirements for specific amino acids (0.79–0.82 for PDCAAS) [doi.org/10.3920/JIFF2018.0048]. Toxicological aspects such as subchronic toxicity of TM diet, including potential hypersensitivity, after orally administration at dose up to 3000 mg/kg/day for 90 days, was already assessed in both sexes of Sprague-Dawley rats [21]. TM was also proven to be non-genotoxic ([51], erratum in [52]).
(DOCX)

**S2 File. Details of animal anaesthesia.** SHR and WKY rats were fully anaesthetized by intra-peritoneal injection of 15 mg/kg Zoletil 100® (Virbac Srl, Milano, Italy) and 4 mg/kg Xylor® (Bio 98, San Lazzaro, Italy). Zoletil 100® is a 1:1 mixture of tiletamine (dissociative anaesthetic acting as NMDA receptor antagonist) and zolazepam (benzodiazepine). Xylor® is composed by the $\alpha_2$ adrenergic receptor agonist xylazine endowed with sedation and muscle relaxation activities.
(DOCX)

**S3 File. RP-HPLC determination of plasma and brain ACE and its inhibitory activity.**
(DOCX)

**S4 File. Liver cytochrome P450- and b5-content, NADPH-cytochrome P450 reductase activity: preparation of liver microsomes.** After the sacrifice the livers were perfused *in situ* through the vena cava with ice-cold physiological saline solution, excised and weighed, chopped, suspended in 4 volumes of 0.25 M sucrose and homogenized in a Potter-Elvejhem homogeniser fitted with a teflon pestle. Homogenates were then centrifuged (10,000 g for 20 min) and the resulting supernatants were further centrifuged at 105,000 g for 1 h. The resulting microsomal pellets were suspended in 0.01 M Tris-HCl buffer, pH 7.6, containing 151 mM KCl, 1 mM EDTA and 20% glycerol, and stored in liquid nitrogen until use.
(DOCX)

**S5 File. Brain homogenate preparation for cytokine assay.** Half brain hemispheres were quickly frozen in liquid nitrogen and stored at -80˚C until analysis. The day of the assay, brains were thawed, homogenated in cold phosphate buffer 0.1 M pH 7.4 (tissue weigh: buffer volume = 1:10) and centrifuged at 10,000 g x 10 min at 4˚C. Afterward the supernatant was collected and used for the ELISA assay, whose sensitivity and assay range were as follows: TNF-$\alpha$ sensitivity $<$ 1 pg/ml, assay range: 7.8–500 pg/ml; IL-6: sensitivity 5.3 pg/ml, assay range 18.8–1200 pg/ml; IL-1ß sensitivity $<$ 12 pg/ml, assay range: 25.6–2500 pg/ml.
(DOCX)

**S6 File. Vascular functionality of aorta rings.** Vascular reactivity was evaluated on aortic rings by acetylcholine and sodium nitroprusside relaxations. Briefly, the thoracic aorta was cleaned of adherent fat and rings 2.5 mm long were cut and placed between stainless-steel

hooks for isometric tension recording in organ chambers. Contractile tension was recorded with a digital PowerLab data acquisition system (PowerLab 8/30; ADInstruments, Castle Hill, Australia) and analysed by using LabChart 7.3.7 Pro (Power Lab; ADInstruments). Rings were stretched to 1 g of tension and equilibrated for 1 h. After pre-contraction with phenylephrine (0.3 μM) the presence of endothelium was verified by the ability of acetylcholine (ACh, 10 μM) to induce relaxation. Concentration-response curves of aortic rings with endothelium to ACh (100 pM–10 μM) and sodium nitroprusside (SNP, 100 pM–100 μM) were performed with and without indomethacin (5 μM; 30 min). In these experiments, the vasodilator responses to ACh and SNP were expressed as percentages of phenylephrine contraction.
(DOCX)

**S7 File. Isolated heart preparation and perfusion.** Spontaneously beating hearts were rapidly explanted and mounted on a Langendorff apparatus for retrograde perfusion *via* the aorta at a constant flow rate of 10 ml/min with a physiological salt solution (PSS) of the following composition (mM): NaCl 118, KCl 4.7, $CaCl_2$ 2.5, $MgSO_4$ 1.2, $NaHCO_3$ 25, $KH_2PO_4$ 1.2, glucose 11.5, Na pyruvate 2, and EDTA 0.5, continuously bubbled with a 95% $O_2$−5% $CO_2$ gas mixture (pH 7.4), and kept at 37˚C [53–55]. The presence of the $Ca^{2+}$ buffer EDTA (0.5 mM) in PSS assures a constant and controlled free $Ca^{2+}$ concentration of 2 mM (EQCAL for Windows software, Biosoft, Cambridge, UK) throughout the experimental session.
(DOCX)

**S8 File. Preparation of brain cortical slices.** The preparation of cortical slices were performed according to standard experimental protocols already reported [18–20]. In particular, the brain was quickly washed in artificial cerebrospinal fluid (ACSF) (composition in mM: 120 NaCl; 2 KCl; 1 $CaCl_2$; 1 $MgSO_4$; 25 HEPES; 1 $KH_2PO_4$; 10 glucose, pH 7.4 and previously bubbled with a 95% $O_2$−5% $CO_2$ gas mixture for 20–30 min) and divided into the two hemispheres. One was used for cytokine assessment, while the other was dissected and cut into 400 μm-thickness cortex slices by using a manual chopper (Stoelting Co., Wood Dale, IL). Slices were transferred into sterile 24-well culture plates containing 0.5 ml ACSF/well previously filtered by passage through a 0.2 μm sterile filter and left at room temperature (25˚C) for 60 min to recover from slicing trauma (equilibration phase). During this period, the medium was removed and replaced with fresh, oxygenated filtered ACSF every 15 min (equilibration phase). After the equilibration phase, oxidative stress was induced by treating the tissue with 10 mM $H_2O_2$ for 1 h [56]. At the end of the treatment, the colorimetric MTT method was used to assess tissue viability as already reported [19]. Slice viability was expressed as a percentage of untreated slices (Basal). Tissue water gain, taken as an index of tissue edema was also assessed and it was calculated by using the formula TE = (wet weight–dry weight)(dry weight)$^{-1}$ and given as g $H_2O$ (g dw)$^{-1}$ [20].
(DOCX)

**S1 Table. Diets composition and ingredients used in the present study.**
(DOCX)

**S2 Table. Effects of the feeding with standard laboratory rodent chow (SD), or SD supplemented with either TM or captopril (C) for 4 weeks on body weight.**
(DOCX)

**S3 Table. Effects of the feeding with standard laboratory rodent chow (SD), or SD supplemented with either TM or captopril (C) for 4 weeks on food intake.**
(DOCX)

**S4 Table. Effects of the feeding for 4 weeks with standard laboratory rodent chow (SD) or SD supplemented with either Tenebrio molitor (TM) or captopril (C) on plasma total low molecular mass thiols (sum of thiol and disulphide forms) in WKY e SHR rats.**
(DOCX)

**S5 Table. Effects of the feeding for 4 weeks with standard laboratory rodent chow (SD) or SD supplemented with either Tenebrio molitor(TM) or captopril (C) on liver cytochrome P450- and b5-content and NADPH-cytochrome P450 reductase activity in WKY e SHR rats.**
(DOCX)

**S1 Fig. Effects of the feeding for 4 weeks with standard laboratory rodent chow (SD) or SD supplemented with either Tenebrio molitor (TM) or captopril (C) on vascular function.** Panel a: 0.3 μM phenylephrine-induced contraction in rat aorta rings. Panel b-d: smooth muscle sensitivity to (b,c) exogenous and (d) endogenous NO: concentration-response curve to (b, c) sodium nitroprusside (SNP) of endothelium-denuded, rat aorta rings or to (d) acetylcholine (ACh) of endothelium-intact rat aorta rings pre-contracted with 0.3 μM phenylephrine. On the ordinate scale, response is reported as mg (panel a) or as percentage of the contraction induced by phenylephrine (phe), taken as 100% (panel b-d). Values are reported as mean ± SEM. *$P<0.05$ vs SHR SD (ANOVA followed by Dunnett post test).
(TIF)

**S2 Fig. Effects of the feeding for 4 weeks with standard laboratory rodent chow (SD) or SD supplemented with either Tenebrio molitor (TM) or captopril (C) on left ventricular pressure (LVDP, panel a) and heart rate (panel b) on isolated heart.** Data are reported as mean ±SEM.
(TIF)

**S3 Fig. Effects of the feeding for 4 weeks with standard laboratory rodent chow (SD) or SD supplemented with either Tenebrio molitor (TM) or captopril (C) IL-6 contents.** Values are reported as mean ± SEM.
(TIF)

## Acknowledgments

The authors wish to thank Jessica Pinassi, Dr. Ettore Vanni and Dr. Ilenia Gaudiosi for the assistance with some of the experiments.

## Author Contributions

**Conceptualization:** Federica Pessina, Maria Frosini, Paola Marcolongo, Fabio Fusi, Simona Saponara, Alessandra Gamberucci, Valeria Francardi, Elena Dreassi.

**Data curation:** Federica Pessina, Maria Frosini, Paola Marcolongo, Fabio Fusi, Simona Saponara, Alessandra Gamberucci, Massimo Valoti, Valeria Francardi, Elena Dreassi.

**Formal analysis:** Federica Pessina, Fabio Fusi, Elena Dreassi.

**Funding acquisition:** Elena Dreassi.

**Investigation:** Maria Frosini, Paola Marcolongo, Fabio Fusi, Massimo Valoti, Daniela Giustarini, Valeria Francardi.

**Methodology:** Federica Pessina, Maria Frosini, Simona Saponara, Alessandra Gamberucci, Massimo Valoti, Daniela Giustarini, Paolo Fiorenzani, Beatrice Gorelli, Valeria Francardi.

**Project administration:** Elena Dreassi.

**Resources:** Maurizio Botta, Elena Dreassi.

**Software:** Maria Frosini.

**Supervision:** Maria Frosini, Simona Saponara, Massimo Valoti, Elena Dreassi.

**Validation:** Maria Frosini, Elena Dreassi.

**Writing – original draft:** Maria Frosini, Fabio Fusi.

**Writing – review & editing:** Federica Pessina, Maria Frosini, Paola Marcolongo, Fabio Fusi, Simona Saponara, Alessandra Gamberucci, Massimo Valoti, Daniela Giustarini, Valeria Francardi, Elena Dreassi.

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
