## [Decision Letter · Decision Letter 0]

13 Jan 2020

PONE-D-19-31063

Antihypertensive, cardio- and neuro-protective effects of Tenebrio molitor (Coleoptera: Tenebrionidae) defatted larvae in spontaneously hypertensive rats

PLOS ONE

Dear FROSINI,

Thank you for submitting your manuscript to PLOS ONE. After careful consideration, we feel that it has merit but does not fully meet PLOS ONE’s publication criteria as it currently stands. Therefore, we invite you to submit a revised version of the manuscript that addresses the points raised during the review process, answering point-by-point the concerns of the reviewers and eventually improving the manuscript with additional experiments if necessary.

We would appreciate receiving your revised manuscript by Feb 27 2020 11:59PM. To enhance the reproducibility of your results, we recommend that if applicable you deposit your laboratory protocols in protocols.io, where a protocol can be assigned its own identifier (DOI) such that it can be cited independently in the future. For instructions see: http://journals.plos.org/plosone/s/submission-guidelines#loc-laboratory-protocols

We look forward to receiving your revised manuscript.

Kind regards,

Luis Eduardo M Quintas, Ph.D.

Academic Editor

PLOS ONE

Journal Requirements:

Reviewers' comments:

Reviewer's Responses to Questions

**Comments to the Author**

1. Is the manuscript technically sound, and do the data support the conclusions?

Reviewer #1: Partly

Reviewer #2: Yes

2. Has the statistical analysis been performed appropriately and rigorously? 

Reviewer #1: Yes

Reviewer #2: Yes

3. Have the authors made all data underlying the findings in their manuscript fully available?

Reviewer #1: Yes

Reviewer #2: Yes

4. Is the manuscript presented in an intelligible fashion and written in standard English?

Reviewer #1: Yes

Reviewer #2: Yes

5. Review Comments to the Author

Reviewer #1: Antihypertensive, cardio- and neuro-protective effects of Tenebrio molitor 2 (Coleoptera: Tenebrionidae) defatted larvae in spontaneously hypertensive rats

Reviewer

The present manuscript studied the effects of a diet enriched with defatted larvae of the mealworm Tenebrio molitor (TM) endowed with ACE inhibitory activity in spontaneously hypertensive rats (SHR) and in the normotensive Wistar Kyoto. This is an important study considering it suggests a dietary approaches to stop hypertension and using the foods to reduce cardiovascular risk factors and improve vascular health.

Comments:

1. Abstract:

Line 38-39: ……Hypertensive rat brain slices were more resistant to oxidative stress and contained lower levels of inflammatory cytokines, while vascular and liver enzyme-activities were not affected…

Authors should inform if this response is from TM group or only for the SHR.

2. Line 38-39:……Hypertensive rat brain slices were more resistant to oxidative stress and contained lower levels of inflammatory cytokines, while vascular and liver enzyme-activities were not affected.

Besides it was demonstrated the TM induced a more resistance to oxidative stress in the brain, this reviewer would like to see its effects at cardiac cells also.

3. Line 40: …..TM can be considered a new functional food that can improve BP in vivo……

Authors should clarify what they want does it means when they say, TM improved blood pressure. It is a general word that does not inform the mechanism or the physiological effect of TM on blood pressure.

Also, when authors inform: line 42: ….and control certain hypertension-associated risk factors…. What authors want to say with…. Certain hypertension-associated risk… it should be more specific.

Introduction

4. Line 44: ……Hypertension is the main risk factor for pathologies such as coronary artery disease……

Pathology is the study of the way diseases and illnesses develop. Authors should avoid use this term as a synonym of disease.

Methods

5. Line 158 : …..Heart contractility was measured as left ventricle pressure (LVP= left ventricular systolic pressure 159 minus left ventricular diastolic pressure)…. I suggest that authors name it as …developed left ventricular systolic pressure, as it is not the absolute systolic pressure, but the systolic minus diastolic pressure.

Also, authors should inform the value of diastolic pressure in the left ventricle, as diastolic pressure is an important determinant of systolic pressure. This is described by Starling Law. That states the systolic pressure increases in response to an increase in the diastolic pressure or diastolic volume, before contraction (the end diastolic volume), when all other factors remain constant. It is important considering the ability of the heart to change its force of contraction in response to changes in diastolic pressure or volume.

Please, inform the diastolic pressure or volume in the hearts.

6. Line 709: Authors informed that nutrition solution for heart perfused at Langendorff preparations had CaCl2 2.5. My question is, usually, rats need a smaller extracellular calcium concentration, around 1.2 to 1.8 mM. Calcium concentration higher them this, usually induces calcium overload and the preparations does not survive longer. I would like to know why authors used this calcium concentration and how long did the hearts survived without demonstrate signal of calcium overload. It is specially complicated when temperature is kept 37 0C.

7. Line 710 to 711: Authors informed that NaHCO3 25 mM, bubbled with a 95% O2–5% CO2 gas mixture (pH 7.4). I would like to know if, at this NaHCO3 concentration, pH was maintained constant during all the experimental time. Did the authors measure the pH in the beginning and in the end of the experiments, or was it estimated? This reviewer would like to see the values. What is the propose of EDTA 0.5 mM?

8. Line 116: why authors measured brain ACE activity and not cardiac? Also, why measure brain IL-6, IL1B and TNP and not vascular iNOS expression to correlate it with those pro-inflammatory mediators?

9. Line 148 to 152: Ex-vivo functional studies and Aorta ring mechanical activity.

Authors used relaxation of phenylephrine-contracted rings to either acetylcholine or sodium nitroprusside as an index of endothelial function and muscle sensitivity to NO. Besides it is an universal and well established way to analyze vasodilatation and endothelial function, isolated, is has less physiological information them when it is used associated with the vasoconstriction phenylephrine dose-response curves activated, to analyze the vascular vasoreactivity. Specially because, the vascular tonus, in the physiological conditions, is regulated by sympathetic system, as the parasimpathetic system does not control aortic function.

Authors should conduct more experiments with the vasoconstriction to phenylephrine using dose-response curves, to analyze the vascular vasoreactivity.

Also, authors should measure de vascular ROS and NO. The reduced NO bioavailability can be attributed to a decreased NO production and/or an increased NO degradation.

A decreased NO production could be due to reduced eNOS expression/presence. The up-regulated expression of eNOS in situations of endothelial dysfunction is likely to be the consequence of an elevated production of hydrogen peroxide, which is a dismutation product of O2·−, and can increase the protein presence of eNOS.

I suggested to use some regular technics (for ex DHE and DAF) and also Western blott for protein expression of iNOS, eNOS and nNOS. Uncoupling of eNOS has been reported in essential hypertensive patients with endothelial dysfunction. So, authors should measure uncoupling eNOS after TM treatment.

Other important enzyme, NADPH oxidase, plays a crucial role in eNOS uncoupling, producing ROS, which leads to oxidation of co-factor BH4 inducing oxidative stress. It should be measured.

Results and Discussion

10. Isolated heart preparation and perfusion was described at methods section, but no results were presented at Results section. Please include it.

11. Lines 295 to 301: …..Pre-hypertension is a warning sign of increased risk of cardiovascular disease and end-organ 296 damage. When not detected early, it can progress to more harmful hypertension. However, blood pressure values of pre-hypertensive patients are not high enough to warrant prescription of antihypertensive drugs. This gap has prompted research into functional foods as a potential source 299 of alternative therapies. In this scenario, food containing peptides that are inactive in the parent protein, but become ACE inhibitors when released during enzymatic digestion or food processing, can play an important role in reducing blood pressure ….

This sentence is not necessary at discussion section as it seems a replay of introduction. I would suggest take it out.

12. Authors inform in the manuscript that: …A feature of particular relevance shown by TM was the marked decrease in blood pressure and hear rate observed in SHRs….

It seems that TM avoided the increment on blood pressure better than to reduce it. In this study. It would be the case if the authors had permitted the BP to increase, them, treated with TM to make it reduce.

Reviewer #2: The manuscript "Antihypertensive, cardio- and neuro-protective effects of Tenebrio molitor (Coleoptera: Tenebrionidae) defatted larvae in spontaneously hypertensive rats” is very well written, and easy to understand. The topic is interesting and important because it deals with potential functional food as a source of alternative therapies for pre-hypertensive patients. Methods section provide enough details for the general reader to repeat the experiments. Аuthors report that diet enriched with defatted larvae of the mealworm in SHR caused a significant reduction of systolic blood pressure, heart rate and coronary perfusion pressure, also increased ACE inhibitory activity in plasma, and decreased inflammatory cytokines IL-1β and TNFα levels in brain homogenates. Further, hypertensive rat brain slices were more resistant to oxidative stress. I suggest that in the future studies focus is on the mechanistic investigations and explaining in more detail the cellular and molecular mechanisms of action of proteins from defatted larvae of the mealworm in hypertensive rats.

6. PLOS authors have the option to publish the peer review history of their article (what does this mean?). If published, this will include your full peer review and any attached files.

Reviewer #1: No

Reviewer #2: No

---

## [Author Response · Author response to Decision Letter 0]

30 Mar 2020

Comments:

1. Abstract: Line 38-39: ……Hypertensive rat brain slices were more resistant to oxidative stress and contained lower levels of inflammatory cytokines, while vascular and liver enzyme-activities were not affected…Authors should inform if this response is from TM group or only for the SHR.

As the sentence reported “hypertensive rat brain slices”, we though that it was clear that data refers to SHR rats. We agree however that the sentence could be misleading, and thus it has been changed into “Rat brain slices of SHR …”.

2. Line 38-39:……Hypertensive rat brain slices were more resistant to oxidative stress and contained lower levels of inflammatory cytokines, while vascular and liver enzyme-activities were not affected. Besides it was demonstrated the TM induced a more resistance to oxidative stress in the brain, this reviewer would like to see its effects at cardiac cells also.

In the present study, hearts of normotensive and hypertensive rats were used to perform the experiments with Langendorff-apparatus. For this reason, it was not possible to assess oxidative stress on cardiac cells immediately after the end of the treatments. The present results, however, encouraged us to deepen the effects of TM in a following in vivo and ex-vivo study, which will be focused on the changes induced at cardiovascular level by the proteins from defatted larvae of the mealworm in hypertensive rats, as also suggested by Referee #2.

3. Line 40: …..TM can be considered a new functional food that can improve BP in vivo……

Authors should clarify what they want does it means when they say, TM improved blood pressure. It is a general word that does not inform the mechanism or the physiological effect of TM on blood pressure. Also, when authors inform: line 42: ….and control certain hypertension-associated risk factors…. What authors want to say with…. Certain hypertension-associated risk… it should be more specific.

We agree with the Referee that the sentence in line 40 is not sufficiently clear. It has been changed as follows:

These results suggest that TM can be considered a new functional food that can lower BP in vivo and thus control cardiovascular-associated risk factors such as hypertension.

Introduction

4. Line 44: ……Hypertension is the main risk factor for pathologies such as coronary artery disease……Pathology is the study of the way diseases and illnesses develop. Authors should avoid use this term as a synonym of disease.

We agree with the Referee and the text has been amended accordingly. 

Methods

5. Line 158 : …..Heart contractility was measured as left ventricle pressure (LVP= left ventricular systolic pressure minus left ventricular diastolic pressure)…. I suggest that authors name it as …developed left ventricular systolic pressure, as it is not the absolute systolic pressure, but the systolic minus diastolic pressure.

We agree with the point raised and thus LVP has been replaced by LVDP in text, Figures and their legend.

Also, authors should inform the value of diastolic pressure in the left ventricle, as diastolic pressure is an important determinant of systolic pressure. This is described by Starling Law. That states the systolic pressure increases in response to an increase in the diastolic pressure or diastolic volume, before contraction (the end diastolic volume), when all other factors remain constant. It is important considering the ability of the heart to change its force of contraction in response to changes in diastolic pressure or volume. Please, inform the diastolic pressure or volume in the hearts.

According to referee’s suggestion we have measured the diastolic pressure (see Figure below). As already reported for LVDP (Supplementary S2 Fig.), neither Tenebrio Molitor (TM), nor captopril (C) affected the LVEDP.

Effects of the feeding for 4 weeks with standard laboratory rodent chow (SD) or SD supplemented with either Tenebrio molitor (TM) or captopril (C) on left ventricular end diastolic pressure (LVEDP) on isolated heart. Data are reported as mean±SEM. 

6. Line 709: Authors informed that nutrition solution for heart perfused at Langendorff preparations had CaCl2 2.5. My question is, usually, rats need a smaller extracellular calcium concentration, around 1.2 to 1.8 mM. Calcium concentration higher them this, usually induces calcium overload and the preparations does not survive longer. I would like to know why authors used this calcium concentration and how long did the hearts survived without demonstrate signal of calcium overload. It is specially complicated when temperature is kept 37 0C.

The 2.5 mM calcium Krebs and Henseleit perfusion solution has always been used worldwide as well as in our laboratories (see for example: Saponara et al, Eur J Pharmacol 2007, 563(1-3):160-3; Gemma et al, J Med Chem 2012, 55(15):6948-67 Fusi et al, J Cardiovasc Pharmacol 2017, 70(6):405-410; Pessina et al, Biochem Pharmacol 2018;155:434-443). Moreover, it is reported in seminal papers (see Skrzypiec-Spring et al (2007) J Pharmacol Toxicol Methods;55:113-26 and chapters of books [Dhein S (2005). The Langendorff Heart in Practical Methods in Cardiovascular Research. Ed Dhein S, Mohr FW, Delmar M. Springer Nature; Doring, HJ and Dehnert H (1987). The isolated perfused heart according to Langendorff. BVM-Biomesstechnic Verlag) describing the Langendorff technique.

In addition, the presence of 0.5 mM EDTA in our solution, ensures a free calcium concentration of 2 mM (EQCAL for Windows software, Biosoft, Cambridge, UK) and we performed experiments to assess the stability of our preparations up to 3 hours from the beginning of the experiment, without any evidence of calcium overload.

7. Line 710 to 711: Authors informed that NaHCO3 25 mM, bubbled with a 95% O2–5% CO2 gas mixture (pH 7.4). I would like to know if, at this NaHCO3 concentration, pH was maintained constant during all the experimental time. Did the authors measure the pH in the beginning and in the end of the experiments, or was it estimated? This reviewer would like to see the values. What is the propose of EDTA 0.5 mM?

The perfusion solution was continuously gassed with 5% CO2 and 95% O2 at a constant temperature of 37°C to maintain an appropriate pH of 7.4. In S7 file is now reported that bubbling was continuous, being this not clearly stated. pH was checked several times, at the beginning, during and at the end of the experimental session, and it was found to be constant. 

EDTA Ca2+ buffering capacity was exploited to ensure a constant and controlled free Ca2+ concentration throughout the experimental session.

8. Line 116: why authors measured brain ACE activity and not cardiac? Also, why measure brain IL-6, IL1B and TNP and not vascular iNOS expression to correlate it with those pro-inflammatory mediators?

Cardiac ACE activity was not assessed, as heart were used to perform the experiments with isolated organ. As previously outlined, however, we are planning a novel in vivo and ex-vivo study, which will be focused on the changes induced at cardiovascular level by the proteins from defatted larvae of the mealworm in hypertensive rats, with particular regard to the role played by vascular NO, iNOS, eNOS and nNOS expression and inflammation (see also the answer to point 9). 

9. Line 148 to 152: Ex-vivo functional studies and Aorta ring mechanical activity.

Authors used relaxation of phenylephrine-contracted rings to either acetylcholine or sodium nitroprusside as an index of endothelial function and muscle sensitivity to NO. Besides it is an universal and well established way to analyze vasodilatation and endothelial function, isolated, is has less physiological information them when it is used associated with the vasoconstriction phenylephrine dose-response curves activated, to analyze the vascular vasoreactivity. Specially because, the vascular tonus, in the physiological conditions, is regulated by sympathetic system, as the parasimpathetic system does not control aortic function. Authors should conduct more experiments with the vasoconstriction to phenylephrine using dose-response curves, to analyze the vascular vasoreactivity. Also, authors should measure de vascular ROS and NO. The reduced NO bioavailability can be attributed to a decreased NO production and/or an increased NO degradation.

A decreased NO production could be due to reduced eNOS expression/presence. The up-regulated expression of eNOS in situations of endothelial dysfunction is likely to be the consequence of an elevated production of hydrogen peroxide, which is a dismutation product of O2·−, and can increase the protein presence of eNOS. I suggested to use some regular technics (for ex DHE and DAF) and also Western blott for protein expression of iNOS, eNOS and nNOS. Uncoupling of eNOS has been reported in essential hypertensive patients with endothelial dysfunction. So, authors should measure uncoupling eNOS after TM treatment. Other important enzyme, NADPH oxidase, plays a crucial role in eNOS uncoupling, producing ROS, which leads to oxidation of co-factor BH4 inducing oxidative stress. It should be measured. 

We thank the Referee for his suggestion. The vasoreactivity to phenylephrine has been analysed and discussed and now is included in the revised Supporting Figure 1, panel a. However, as TM supplementation did not improve either endothelium-dependent relaxation or smooth muscle sensitivity to phenylephrine, we believe that an in depth analysis of eNOS/NADPH oxidase expression as well as ROS production in aorta rings would not help clarifying the mechanism(s) underpinning the beneficial effect of TM treatment on systemic blood pressure and, therefore, has not been executed.

Results and Discussion 10. Isolated heart preparation and perfusion was described at methods section, but no results were presented at Results section. Please include it.

In the previous version of the manuscript Langendorff results were already reported. CPP results were reported in Fig 2d and LVDP and BPM in Suppl Fig 2 (wrongly linked to Suppl Fig 3, now properly amended).

In paragraph “Effects of TM on cardiovascular mechanics” was already stated “Regarding heart function, no type of diet fed to WKY or SHRs affected LVDP or heart rate as evaluated by the Langendorff perfused heart model. SHRs fed the standard diet showed higher coronary perfusion pressure than did the matched normotensive strain, with both captopril- and TM -supplemented diets bringing this parameter back to control values (Fig 2, panel d).

11. Lines 295 to 301: …..Pre-hypertension is a warning sign of increased risk of cardiovascular disease and end-organ 296 damage. When not detected early, it can progress to more harmful hypertension. However, blood pressure values of pre-hypertensive patients are not high enough to warrant prescription of antihypertensive drugs. This gap has prompted research into functional foods as a potential source 299 of alternative therapies. In this scenario, food containing peptides that are inactive in the parent protein, but become ACE inhibitors when released during enzymatic digestion or food processing, can play an important role in reducing blood pressure ….This sentence is not necessary at discussion section as it seems a replay of introduction. I would suggest take it out.

The suggestion has been taken and the text omitted. 

12. Authors inform in the manuscript that: …A feature of particular relevance shown by TM was the marked decrease in blood pressure and heart rate observed in SHRs….

It seems that TM avoided the increment on blood pressure better than to reduce it. In this study. It would be the case if the authors had permitted the BP to increase, them, treated with TM to make it reduce.

We agree with the Referee that the original sentence might be ambiguous, and thus it was modified accordingly as reported below:

A feature of particular relevance shown by TM was its ability to prevent the marked increase in blood pressure and heart rate observed in SHRs, known to be associated with vascular and heart protection. 

Reviewer #2: The manuscript "Antihypertensive, cardio- and neuro-protective effects of Tenebrio molitor (Coleoptera: Tenebrionidae) defatted larvae in spontaneously hypertensive rats” is very well written, and easy to understand. The topic is interesting and important because it deals with potential functional food as a source of alternative therapies for pre-hypertensive patients. Methods section provide enough details for the general reader to repeat the experiments. Аuthors report that diet enriched with defatted larvae of the mealworm in SHR caused a significant reduction of systolic blood pressure, heart rate and coronary perfusion pressure, also increased ACE inhibitory activity in plasma, and decreased inflammatory cytokines IL-1β and TNFα levels in brain homogenates. Further, hypertensive rat brain slices were more resistant to oxidative stress. I suggest that in the future studies focus is on the mechanistic investigations and explaining in more detail the cellular and molecular mechanisms of action of proteins from defatted larvae of the mealworm in hypertensive rats.

We thank the Referee for the suggestion that will be followed in the next study focused on the mechanisms responsible for the effects that the proteins from defatted larvae of the mealworm exert at cardiovascular level in hypertensive rats.

---

## [Decision Letter · Decision Letter 1]

27 Apr 2020

PONE-D-19-31063R1

Antihypertensive, cardio- and neuro-protective effects of Tenebrio molitor (Coleoptera: Tenebrionidae) defatted larvae in spontaneously hypertensive rats

PLOS ONE

Dear FROSINI,

Thank you for submitting your manuscript to PLOS ONE. After careful consideration, we feel that it has merit but does not fully meet PLOS ONE’s publication criteria as it currently stands. Therefore, we invite you to submit a revised version of the manuscript that addresses the points raised during the review process.

Specifically, reviewer #1 is still concerned about EDTA addition and the concentration of calcium in Langendorff experiments. Thus, I would require that you insert in the text (in supporting information S7) the explanation you presented to the reviewer (i.e., why EDTA was added, how free calcium was estimated and some literature reference about 2.5 mM CaCl2) in order to clear to the reader. 

We would appreciate receiving your revised manuscript by Jun 11 2020 11:59PM. To enhance the reproducibility of your results, we recommend that if applicable you deposit your laboratory protocols in protocols.io, where a protocol can be assigned its own identifier (DOI) such that it can be cited independently in the future. For instructions see: http://journals.plos.org/plosone/s/submission-guidelines#loc-laboratory-protocols

We look forward to receiving your revised manuscript.

Kind regards,

Luis Eduardo M Quintas, Ph.D.

Academic Editor

PLOS ONE

Reviewers' comments:

Reviewer's Responses to Questions

**Comments to the Author**

1. If the authors have adequately addressed your comments raised in a previous round of review and you feel that this manuscript is now acceptable for publication, you may indicate that here to bypass the “Comments to the Author” section, enter your conflict of interest statement in the “Confidential to Editor” section, and submit your "Accept" recommendation.

Reviewer #1: (No Response)

2. Is the manuscript technically sound, and do the data support the conclusions?

Reviewer #1: Yes

3. Has the statistical analysis been performed appropriately and rigorously? 

Reviewer #1: Yes

4. Have the authors made all data underlying the findings in their manuscript fully available?

Reviewer #1: Yes

5. Is the manuscript presented in an intelligible fashion and written in standard English?

Reviewer #1: Yes

6. Review Comments to the Author

Reviewer #1: Authors answer:

In addition, the presence of 0.5 mM EDTA in our solution, ensures a free calcium

concentration of 2 mM (EQCAL for Windows software, Biosoft, Cambridge, UK) and we

performed experiments to assess the stability of our preparations up to 3 hours from

the beginning of the experiment, without any evidence of calcium overload.

7. Line 710 to 711: Authors informed that NaHCO3 25 mM, bubbled with a 95%

O2–5% CO2 gas mixture (pH 7.4). I would like to know if, at this NaHCO3

concentration, pH was maintained constant during all the experimental time. Did the

authors measure the pH in the beginning and in the end of the experiments, or was it

estimated? This reviewer would like to see the values. What is the propose of EDTA

0.5 mM?

EDTA Ca2+ buffering capacity was exploited to ensure a constant and controlled free

Ca2+ concentration throughout the experimental session.

Reviewer question:

It still not clear why authos used EDTA 0.5 mM in order to buffer high Ca++ concentration .

If the idea was to use Calcium 2.0 mM, why did authors used a high concentration (2.5 mM), and them used EDTA to reduce it, as most of paper have been using extracecullar calcium in rat heart 1.2 to 1.8 mM.

7. PLOS authors have the option to publish the peer review history of their article (what does this mean?). If published, this will include your full peer review and any attached files.

Reviewer #1: No

---

## [Author Response · Author response to Decision Letter 1]

1 May 2020

Reviewer #1 question:

It still not clear why authors used EDTA 0.5 mM in order to buffer high Ca++ concentration .

If the idea was to use Calcium 2.0 mM, why did authors used a high concentration (2.5 mM), and them used EDTA to reduce it, as most of paper have been using extracecullar calcium in rat heart 1.2 to 1.8 mM.

As suggested by the Editor, in supporting information S7 the explanation about why EDTA was added, how free calcium was estimated were insert along with appropriate references (i.e. 53-55) about 2.5 mM CaCl2 in order to clear to the reader.

---

## [Editor Report · Decision Letter 2]

13 May 2020

Antihypertensive, cardio- and neuro-protective effects of Tenebrio molitor (Coleoptera: Tenebrionidae) defatted larvae in spontaneously hypertensive rats

PONE-D-19-31063R2

Dear Dr. FROSINI,

We are pleased to inform you that your manuscript has been judged scientifically suitable for publication and will be formally accepted for publication once it complies with all outstanding technical requirements.

With kind regards,

Luis Eduardo M Quintas, Ph.D.

Academic Editor

PLOS ONE
---

## [Editor Report · Acceptance letter]

15 May 2020

PONE-D-19-31063R2 

Antihypertensive, cardio- and neuro-protective effects of Tenebrio molitor (Coleoptera: Tenebrionidae) defatted larvae in spontaneously hypertensive rats 

Dear Dr. Frosini:

I am pleased to inform you that your manuscript has been deemed suitable for publication in PLOS ONE. Congratulations! Your manuscript is now with our production department. 

With kind regards,

on behalf of

Dr. Luis Eduardo M Quintas 

Academic Editor

PLOS ONE